# Training-Free Feature Reconstruction with Sparse Optimization for Vision-Language Models

Yi Zhang
Harbin Institute of Technology,
Southern University of Science and
Technology
Shenzhen, China
zhangyi2021@mail.sustech.edu.cn

Ke Yu
University of California San Diego
San Diego, CA
key022@ucsd.edu

Angelica I Aviles-Rivero
University of Cambridge
Cambridge, United Kingdom
ai323@cam.ac.uk

Jiyuan Jia
Southern University of Science and
Technology
Shenzhen, China
jiajy2018@mail.sustech.edu.cn

Yushun Tang
Southern University of Science and
Technology
Shenzhen, China
tangys2022@mail.sustech.edu.cn

Zhihai He*
Southern University of Science and
Technology
Shenzhen, China
hezh@sustech.edu.cn

## Abstract

In this paper, we address the challenge of adapting vision-language models (VLMs) to few-shot image recognition in a training-free manner. We observe that existing methods are not able to effectively characterize the semantic relationship between support and query samples in a training-free setting. We recognize that, in the semantic feature space, the feature of the query image is a linear and sparse combination of support image features since support-query pairs are from the class and share the same small set of distinctive visual attributes. Motivated by this interesting observation, we propose a novel method called *Training-free Feature ReConstruction with Sparse optimization* (**TaCo**), which formulates the few-shot image recognition task as a feature reconstruction and sparse optimization problem. Specifically, we exploit the VLM to encode the query and support images into features. We utilize sparse optimization to reconstruct the query feature from the corresponding support features. The feature reconstruction error is then used to define the reconstruction similarity. Coupled with the text-image similarity provided by the VLM, our reconstruction similarity analysis accurately characterizes the relationship between support and query images. This results in significantly improved performance in few-shot image recognition. Our extensive experimental results on few-shot recognition demonstrate that our method outperforms existing state-of-the-art approaches by substantial margins.

## CCS Concepts

• **Computing methodologies** → **Computer vision**; **Natural language processing**.

---

*Corresponding author.

## Keywords

Vision-Language, Generalization, Few-Shot Learning

**ACM Reference Format:**
Yi Zhang, Ke Yu, Angelica I Aviles-Rivero, Jiyuan Jia, Yushun Tang, and Zhihai He. 2024. Training-Free Feature Reconstruction with Sparse Optimization for Vision-Language Models. In *Proceedings of the 32nd ACM International Conference on Multimedia (MM '24), October 28-November 1, 2024, Melbourne, VIC, Australia.* ACM, New York, NY, USA, 10 pages. https://doi.org/10.1145/3664647.3680710

## 1 Introduction

Recently, considerable attention has been directed towards large-scale pre-trained Vision-Language Models (VLMs) for natural language processing and computer vision. These models exploit extensive datasets containing both images and corresponding textual descriptions to acquire unified representations of visual and textual data. VLMs, such as CLIP [36], leverage extensive pre-training to establish connections between text and images, showcasing notable achievements in few-shot learning through fine-tuning [12, 36]. Existing fine-tuning methods for few-shot image recognition can be classified into two categories, (1) input-level prompting approaches, such as CoOp [61], CoCoOp [60], ProDA [30], and PLOT [2], and (2) feature-level fine-tuning methods, such as CLIP-Adapter [12] and Tip-Adapter [56]. For example, the CoOp method [61] introduces learnable prompts aimed at distilling task-specific knowledge. PLOT [2] learns multiple comprehensive prompts to depict diverse category characteristics. CLIP-Adapter [12] learns a feature adapter to enhance conventional fine-tuning outcomes.

Among existing methods, training-free, few-shot image recognition based on VLMs has emerged as an interesting research task. Tip-Adapter [56], following the footsteps of CLIP-Adapter, presents a training-free paradigm by establishing a key-value cache model from few-shot samples. The APE method [63] analyzes the inter-class disparity in the downstream data and decouple the domain-specific knowledge from the CLIP-extracted cache model for training-free few-shot image recognition.

We observe that existing training-free methods, mainly based on nearest neighbor analysis, are not able to effectively characterize the sophisticated semantic relationship between support and query

Query Images

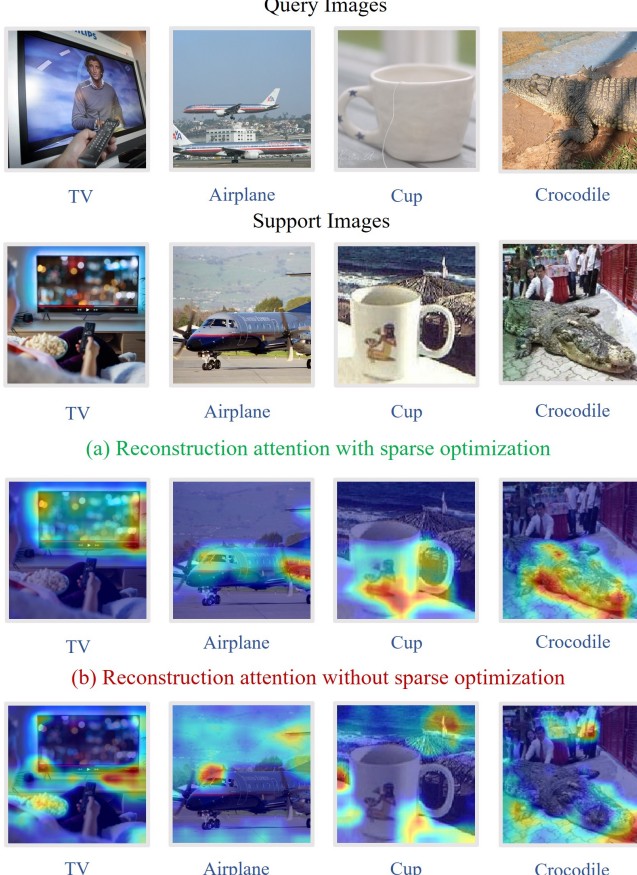

TV          Airplane          Cup          Crocodile

Support Images

TV          Airplane          Cup          Crocodile

(a) Reconstruction attention with sparse optimization

TV          Airplane          Cup          Crocodile

(b) Reconstruction attention without sparse optimization

TV          Airplane          Cup          Crocodile

**Figure 1: Feature reconstruction attention map. (a) shows feature reconstruction with sparse optimization, and (b) for feature reconstruction without sparse optimization. As shown in (a) and (b), sparse optimization can direct attention toward more informative target features and minimize attention to the profitless features for reconstruction.**

samples. In this paper, we recognize that the query and support images of the same class share a common small set of distinctive visual attributes. For example, the query and support images from the "cat" class share the same distinctive visual features of cats, such as cat eyes, mouth, paws, and tail. Motivated by this observation, we hypothesize that, in the semantic feature space, the feature of the query image can be considered as a linear and sparse combination of support image features since the support-query pairs share the same small set of visual attributes.

Based on this hypothesis, we propose a novel method called *Training-free Feature ReConstruction with Sparse optimization* (**TaCo**), formulating the few-shot image recognition task as a feature reconstruction and sparse optimization problem. Specifically, using the VLM, we encode the query and support images into features. We attempt to leverage sparse optimization to reconstruct the query feature from the corresponding support features. As illustrated in Figure 1, the proposed feature reconstruction between the query

and support images using sparse optimization is able to guide attention toward more informative target regions. This approach minimizes attention to the features that hold minimal significance in the reconstruction process. If the query image is from the same class as the support images, their reconstruction error should be small. Therefore, we can use the feature reconstruction error to define the reconstruction similarity for few-shot image recognition. Coupled with the text-image similarity provided by the VLM, reconstruction similarity analysis can accurately characterize the relationship between support and query images, thereby resulting in significantly improved performance in few-shot image recognition.

The contributions of this work can be summarized as follows: 1) We propose a novel training-free method for adapting VLMs to few-shot image recognition by image feature reconstruction with spare optimization. 2) We develop a method to solve the sparse optimization problem for query feature reconstruction based on alternative direction methods. We fuse the feature prediction similarity obtained from this reconstruction process and the text-image similarity obtained by the CLIP model to form few-shot image recognition. 3) Our extensive experimental results demonstrate that the proposed method has significantly improved the performance of training-free few-shot image recognition, and outperforms existing state-of-the-art approaches by substantial margins.

## 2 Related Work

### 2.1 Pre-Trained Vision-Language Models

VLMs establish connections between image content and language. Numerous studies have delved into VLMs to acquire comprehensive visual representations guided by natural language supervision [7, 14, 25, 39]. Recently, VLMs based on contrastive learning have demonstrated remarkable performance by leveraging large-scale, noisy image-text pairs from the web. For example, CLIP [36] and ALIGN [20] employ contrastive loss to learn aligned representations of image and text, pulling close the representations of matching pairs and pushing apart those of mismatched pairs. Guided by natural language supervision, these VLMs not only acquire robust visual representations but also exhibit seamless transferability to diverse downstream tasks, encompassing image retrieval [10, 29], visual grounding [26, 52], visual question answering [10, 24, 62], as well as image manipulation and synthesis [19, 22, 40, 54].

### 2.2 Adapting VLMs to Few-Shot Classification

Enhancing the adaptability of VLMs to few-shot classification is achievable through fine-tuning. Current methods fall into two categories: input-level prompting and feature-level adapters.

**Input-level Prompting Methods** are influenced by the success observed in prefix-tuning within the realm of natural language processing [6, 13, 21, 28]. These methods, tailored for fine-tuning pre-trained VLMs, center their efforts on crafting thoughtful prompts and introducing adaptable context to distill task-specific information from the encoded knowledge [43, 60, 61]. Recent advancements in prompt tuning methods that have demonstrated substantial enhancements include CoOp [61], a groundbreaking work that optimizes prompt context using learnable vectors in a unified or class-specific manner. Additionally, CoCoOp [60] builds

upon CoOp by incorporating the ability to generate vectors conditioned on each image, addressing the challenge of generalizing to unseen classes. TPT [32] dynamically learns adaptive prompts with just a single test sample, while ProDA [30] captures diverse prompt distributions to accommodate varying visual representations. DeFo [48] leverages feature-level textual prompts to learn decomposed visual features. PLOT [2] employs the strategy of learning multiple comprehensive prompts to describe diverse category characteristics. In addition, CPL [58] exploits the powerful comprehension of VLMs and utilizes visual concepts to further improve benchmark performance.

**Feature-level Adapter Methods** directly adjust the representations generated by CLIP's visual and text encoders. Taskres [53] operates directly on the text-based classifier, explicitly separating prior knowledge from pre-trained models and new knowledge relevant to a target task. Pioneering this approach, CLIP-Adapter [12] introduces an additional feature adapter to enhance conventional fine-tuning outcomes. Subsequently, Tip-Adapter [56] achieves further improvements by constructing a key-value cache model based on low-shot samples and fine-tuning for a reduced number of epochs. BDC-Adapter [57], leverages the Brownian Distance Covariance to better model both linear and nonlinear relations, to achieve better reasoning ability. Following the adapter-based paradigm, our work adapts VLMs to few-shot classification by feature reconstruction.

## 2.3 Reconstruction-Related Few-Shot Learning

Feature reconstruction, a well-established technique in object tracking and alignment [9, 42, 47], has recently found application in few-shot image classification. DeepEMD [55] addresses reconstruction as an optimal transport problem. CrossTransformer [8] and CrossAttention [18] incorporate attention modules projecting query features into the support feature space. They compare class-conditioned projections to the target, predicting class membership. FRN [50] frames membership as a feature map reconstruction problem by regressing directly from support features to query features in closed form. In this work, we propose a parameter-efficient reconstruction-based method for adapting VLMs to few-shot learning.

## 3 Proposed Method

### 3.1 Method Overview

As shown in Figure 2, given the pre-trained CLIP and a new dataset with $N$-shot $D$-class training samples for few-shot learning, there are $N$ annotated images in each of the $D$ categories. For each class $d \in D$, using the CLIP image encoder, we encode support images and pool their features into a feature matrix denoted as $\mathbf{S}_d \in \mathbb{R}^{NH_4W_4 \times C}$, referred as the support feature map. Similarly, we generate the query feature map $\mathbf{m}_q \in \mathbb{R}^{H_4W_4 \times C}$ for the query image $x_q$. Then, we attempt to reconstruct the feature map $\mathbf{m}_q$ through a weighted combination of the rows within $\mathbf{S}_d$. The reconstructed query feature map can be calculated by $\mathbf{m}_q^* = \mathbf{w}\mathbf{S}_d$. Here, $\mathbf{w} \in \mathbb{R}^{H_4W_4 \times NH_4W_4}$ is optimized such that the product $\mathbf{w}\mathbf{S}_d$ closely approximates $\mathbf{m}_q$, which we formulate as a sparse optimization problem. In other words, we require that the reconstruction matrix $\mathbf{w}$ be sparse so that the query feature can be reconstructed from a small set of selected features in the support feature map. From the attention perspective, we wish that the query image can inherit the

distinctive visual attributes of the support set and the reconstruction process can guide the attention towards this small set of visual attributes. If the query image is from the same class as the support images, their reconstruction error should be small. Therefore, we can use the feature reconstruction error to define the reconstruction similarity for few-shot image recognition. Also, we analyze the similarity between the query image feature and the text feature of each class. The reconstruction similarity and the text-image similarity are then fused to form the final class prediction.

Our model is constructed based on CLIP, utilizing $E_t$ as the text encoder and $E_v$ as the image encoder [37]. For instance, considering the ResNet encoder, which consists of a total of 4 stages, we denote the feature maps as $\{\mathbf{x}_i\}_{i=1}^4$. In contrast to the original ResNet, CLIP introduces a slight modification by incorporating an attention-pooling layer. CLIP initially applies global average pooling to $\mathbf{x}_4 \in \mathbb{R}^{H_4W_4 \times C}$ to derive a global feature $\bar{\mathbf{x}}_4 \in \mathbb{R}^{1 \times C}$, where $H_4$, $W_4$, and $C$ represent the height, width, and number of channels of the 4th stage feature maps in the backbone. The combined features $[\bar{\mathbf{x}}_4, \mathbf{x}_4]$ are subsequently inputted into a multi-head self-attention layer (MHSA), represented as $[\bar{\mathbf{m}}, \mathbf{m}] = \text{MHSA}([\bar{\mathbf{x}}_4, \mathbf{x}_4])$. In the conventional CLIP training process, the global feature $\bar{\mathbf{m}}$ serves as the image encoder output, while other outputs $\mathbf{m}$ are typically disregarded. However, we have found an intriguing aspect of $\mathbf{m}$: it retains sufficient spatial information and can function as a feature map. Furthermore, it should be mentioned that in architectures such as ViT, obtaining $\mathbf{m}$ can be achieved similarly by omitting the class token from the output.

## 3.2 Sparsity Guided Feature Reconstruction

In this section, we discuss why sparse optimization is beneficial for reconstructing features. For a better explanation, we denote two sets: $C$ containing only cat images and $\mathcal{D}$ containing only dog images. The VLM is represented as $\Phi$, which takes an image $\mathbf{I}$ as input and produces $N$ feature vectors for each class. The set of extracted feature vectors serving as a basis for the cat set $C$ is denoted as $\mathcal{S}_C = \{\mathbf{c}_1, \mathbf{c}_2, \ldots, \mathbf{c}_{N_c}\}$, where $N_c \leq N$ is the cardinality of $\mathcal{S}_C$ and feature basis are independent to each other and every feature vector of a pure cat image. Similarly, for the dog set $\mathcal{D}$, we have $\mathcal{S}_{\mathcal{D}} = \{\mathbf{d}_1, \mathbf{d}_2, \ldots, \mathbf{d}_{N_d}\}$. Let $\mathbf{C} = [\mathbf{c}_1, \mathbf{c}_2, \ldots, \mathbf{c}_{N_c}]$ and $\mathbf{D} = [\mathbf{d}_1, \mathbf{d}_2, \ldots, \mathbf{d}_{N_d}]$.

**(1) Ensuring Sparsity to Prevent Misconstruction.** Consider the representation $\mathbf{v}_t = \mathbf{C}\mathbf{w}_t + \mathbf{n}_t$, where $\mathbf{w}_t$ is a sparse vector. Additionally, $\mathbf{n}_t$ represents a deviation orthogonal to all feature vectors in $\mathcal{S}_C$. However, some dog feature vectors may be highly correlated with those of cats. For simplicity, let's consider the linear correlation $[\mathbf{c}_1, \mathbf{c}_2, \ldots, \mathbf{c}_{N_t}] = [\mathbf{d}_1, \mathbf{d}_2, \ldots, \mathbf{d}_{N_t}]\mathbf{T}$, where $N_t < \min\{N_c, N_d\}$ and $\mathbf{T} \in \mathbb{R}^{N_t \times N_t}$ is an invertible matrix.

The reconstruction error using the cat feature basis is $\|\mathbf{n}_t\|_2^2$. Yet, we can find a vector $\mathbf{w}_d$ within the dog feature space that yields a comparable reconstruction error. To explore this, we partition $\mathbf{C}$ into $\mathbf{C} = [\mathbf{C}_l, \mathbf{C}_r]$, with $\mathbf{C}_l = [\mathbf{c}_1, \mathbf{c}_2, \ldots, \mathbf{c}_{N_t}]$. We similarly partition $\mathbf{D}$ into $\mathbf{D} = [\mathbf{D}_l, \mathbf{D}_r]$. Additionally, we decompose $\mathbf{w}_t = [\mathbf{w}_l^\top, \mathbf{w}_r^\top]^\top$, where $\mathbf{w}_l = [w_1, w_2, \ldots, w_{N_t}]^\top$. By defining $\mathbf{w}_d = [(\mathbf{T}\mathbf{w}_l)^\top, \mathbf{0}^\top]^\top + \mathbf{u}^\star$, we can achieve a reconstruction error of $\|\mathbf{C}_r\mathbf{w}_r + \mathbf{n}_t - \mathbf{D}\mathbf{u}^\star\|_2^2$. Since $\mathbf{n}_t$ is not orthogonal to the dog feature basis, some parts of it can be canceled. By selecting $\mathbf{u}^\star = \arg\min_{\mathbf{u}} \|\mathbf{C}_r\mathbf{w}_r + \mathbf{n}_t - \mathbf{D}\mathbf{u}\|_2^2$,

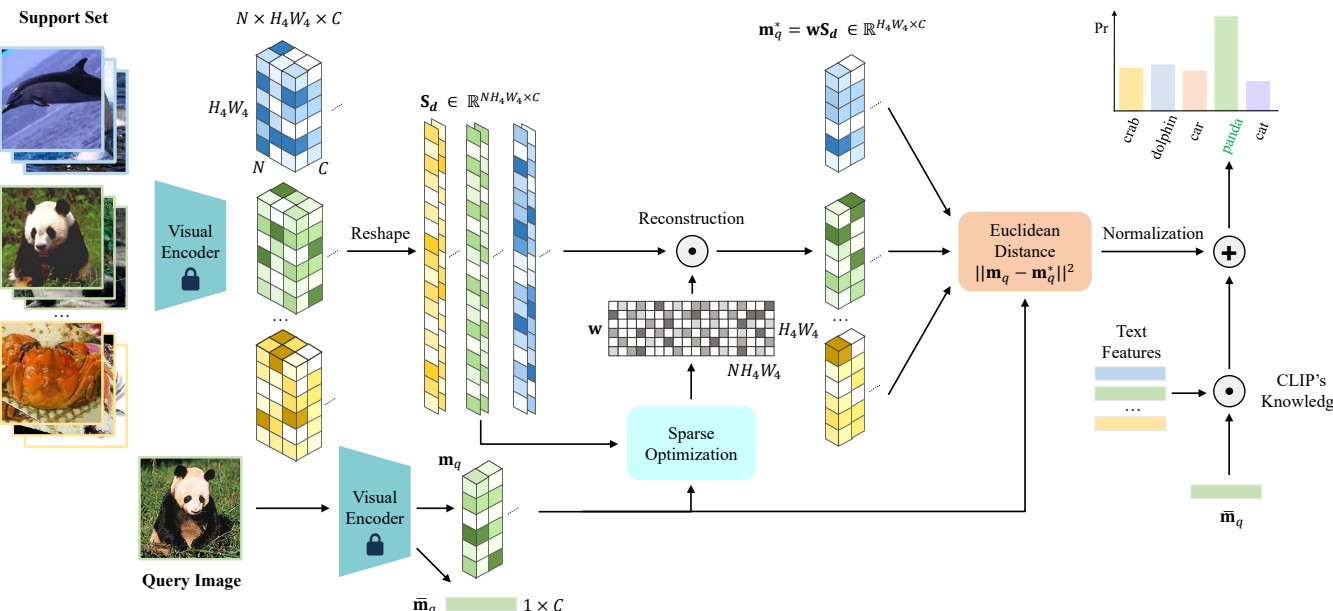

**Figure 2: An overview of our proposed method for $D$-way $N$-shot classification. We first utilize CLIP's visual encoder to generate feature maps for the support set and query image. Then, we attempt to use the feature maps from the support set of each class to reconstruct the feature map of the query image and utilize the feature reconstruction error as the reconstruction similarity. Therefore, we calculate the cosine similarity between the query image and the text feature of each class as CLIP's text-image similarity. The two similarity scores are then fused together to form the final class prediction. Meanwhile, during the reconstruction, sparse optimization is applied to w to optimize the transformation process. Here, $C$ represents the number of channels, and $H_4$, $W_4$ denote the size of the feature map, respectively.**

it is feasible to construct a $\mathbf{w}_d$ with a small reconstruction error. However, constructing $\mathbf{w}_d$ in this manner will not result in sparsity unless $\mathbf{T}$ is a permutation matrix and $\mathbf{u}^\star$ is sparse. This leads us to leverage the sparsity of $\mathbf{w}_t$.

**(2) Sparsity Enhances Reconstruction Emphasis on Principal Components.** Consider a different scenario: suppose $\mathbf{I}_t$ is a cat image with a dog in the background, as illustrated in Figure 3, where the cat constitutes the majority of the image. Therefore, the image can be represented as:

$$\mathbf{v}_t = \mathbf{C}\mathbf{w}_c + \mathbf{D}\mathbf{w}_d + \mathbf{n}_t = \begin{bmatrix} \mathbf{C} & \mathbf{D} \end{bmatrix} \begin{bmatrix} \mathbf{w}_c \\ \mathbf{w}_d \end{bmatrix} + \mathbf{n}_t = \mathbf{S}\mathbf{w} + \mathbf{n}_t, \quad (1)$$

where $\mathbf{n}_t$ is a deviation orthogonal to the bases of both cat and dog features, $\mathbf{w}_c$ and $\mathbf{w}_d$ are both sparse, and $\mathbf{w}_d$ has a very small norm. Here, $\mathbf{S} = [\mathbf{C}, \mathbf{D}]$ and $\mathbf{w} = [\mathbf{w}_c^\top, \mathbf{w}_d^\top]^\top$.

If $\mathbf{w}_d$ has a small norm, then $\|\mathbf{D}\mathbf{w}_d + \mathbf{n}_t\|_2^2 \leq \|\mathbf{D}\mathbf{w}_d\|_2^2 + \|\mathbf{n}_t\|_2^2$ is small. Under a tolerable reconstruction error threshold, this quantity is smaller than the threshold, we can sparsify $\mathbf{w}$ to the greatest extent possible, making it possible to neglect the contribution of the image from $\mathbf{w}_d$. This implies that we can neglect the interference from the dog and concentrate on the cat's features.

**(3) Formulating Principal Reconstruction as a Sparse Optimization Problem.** In line with the aforementioned concepts, we now cast our reconstruction problem as follows:

$$\mathcal{P}_0: \quad \min_{\mathbf{w}} \|\mathbf{w}\|_0 \ \text{ s.t. } \|\mathbf{m}_\mathbf{q} - \mathbf{S}_d\mathbf{w}\|_2^2 \leq \epsilon. \quad (2)$$

By solving $\mathcal{P}_0$, we aim to minimize the sparsity of the reconstruction coefficient $\mathbf{w}$ under the constraint that the reconstruction error, measured by $\|\mathbf{m}_\mathbf{q} - \mathbf{S}_d\mathbf{w}\|_2^2$, is kept below a specified threshold $\epsilon$.

### 3.3 Sparse Optimization with ADM

The problem $\mathcal{P}_0$ is widely acknowledged as NP-hard. To tackle this complexity, a common approach is to relax the $l_0$ norm to an $l_p$ norm, where $p$ is a non-zero parameter. In this study, we opt for $p = 1$, a prevalent choice in addressing sparse representation problems. This leads to a relaxed optimization problem:

$$\mathcal{P}_1: \quad \min_{\mathbf{w}} \|\mathbf{w}\|_1 \ \text{ s.t. } \|\mathbf{m}_\mathbf{q} - \mathbf{S}_d\mathbf{w}\|_2^2 \leq \epsilon. \quad (3)$$

According to the Lagrange multiplier theorem, there exists a suitable constant $\lambda$ rendering problem $\mathcal{P}_1$ equivalent to the following unconstrained minimization problem, where $\lambda$ is associated with a very small $\epsilon$:

$$\mathcal{P}_2: \quad \min_{\mathbf{w}} \|\mathbf{w}\|_1 + \lambda \|\mathbf{m}_\mathbf{q} - \mathbf{S}_d\mathbf{w}\|_2^2. \quad (4)$$

The introduction of the $l_1$ norm in the objective function of $\mathbf{P}_2$ renders it a nonsmooth optimization problem. Common optimization algorithms such as the gradient descent algorithm or Newton's method can be employed to solve this problem. However, a challenge arises in selecting an appropriate step size. When some entries of the optimal solution are close to zero, the solution may oscillate around zero, impeding effective convergence if a small step size is

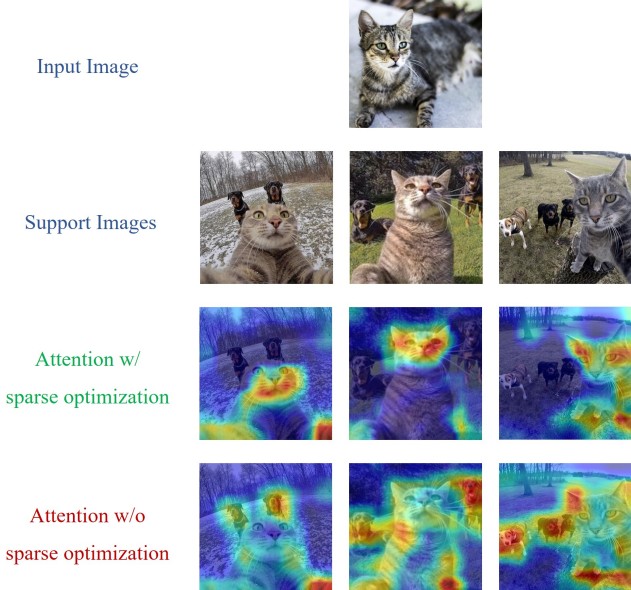

Input Image

Support Images

Attention w/ sparse optimization

Attention w/o sparse optimization

**Figure 3: Sample reconstruction attention images, including different cat images with dogs in the background. We use support images to reconstruct the input image. The visualization results show that sparsity enhances reconstruction emphasis on principal components.**

not employed. This sluggish optimization process emphasizes the need for careful consideration in choosing the step size.

A more realistic approach to handle nonsmoothness is the Alternating Direction Method (ADM). This method introduces an auxiliary variable, optimizing progressively and mutually alongside the original variable, without relying on a specific step size [33]. ADM typically achieves effective convergence in approximately 10 iterations, proving to be more efficient than other methods when dealing with nonsmooth problems like $\mathcal{P}_2$. To solve $\mathcal{P}_2$ using ADM, we first introduce an auxiliary variable to formulate an equivalent problem of $\mathcal{P}_2$.

$$\mathcal{P}_3: \quad \min_{\mathbf{w}} \|\mathbf{w}\|_1 + \lambda \|\mathbf{z}\|_2^2 \quad s.t. \ \mathbf{z} = \mathbf{m_q} - S_d\mathbf{w}. \tag{5}$$

The augmented Lagrangian dual optimization problem of $\mathcal{P}_3$ can be expressed as

$$\mathcal{P}_4: \quad \min_{\mathbf{w},\mathbf{z},\boldsymbol{\mu}} L(\mathbf{w}, \mathbf{z}, \boldsymbol{\mu}) = \|\mathbf{w}\|_1 + \lambda\|\mathbf{z}\|_2^2 + \boldsymbol{\mu}^\top(\mathbf{z} - \mathbf{m_q} + S_d\mathbf{w}) \\ + \frac{\nu}{2}\|\mathbf{z} - \mathbf{m_q} + S_d\mathbf{w}\|_2^2. \tag{6}$$

Here, $\boldsymbol{\mu} \in \mathbb{R}^{d \times 1}$ is the Lagrange multiplier vector, and $\nu$ is the penalty factor. The ADM is employed to solve problem $\mathcal{P}_4$ through the following iterative steps:

$$\mathbf{w}_{k+1} = \underset{\mathbf{w}}{\text{argmin}} \ L(\mathbf{w}, \mathbf{z}_k, \boldsymbol{\mu}_k), \tag{7}$$

$$\mathbf{z}_{k+1} = \underset{\mathbf{z}}{\text{argmin}} \ L(\mathbf{w}_{k+1}, \mathbf{z}, \boldsymbol{\mu}_k), \tag{8}$$

$$\boldsymbol{\mu}_{k+1} = \boldsymbol{\mu}_k - \nu(\mathbf{z}_{k+1} - \mathbf{m_q} + S_d\mathbf{w}_{k+1}). \tag{9}$$

Equation (7) can be expressed as

$$\mathbf{w}_{k+1} = \underset{\mathbf{w}}{\text{argmin}} \ \|\mathbf{w}\|_1 + \lambda\|\mathbf{z}_k\|_2 - \boldsymbol{\mu}_k^\top(\mathbf{z}_k - \mathbf{m_q} + S_d\mathbf{w}) \\ + \frac{\nu}{2}\|\mathbf{z}_k - \mathbf{m_q} + S_d\mathbf{w}\|_2^2 \\ = \underset{\mathbf{w}}{\text{argmin}} \ \|\mathbf{w}\|_1 + \frac{\nu}{2}\|\mathbf{z}_k - \mathbf{m_q} + S_d\mathbf{w} - \frac{\boldsymbol{\mu}_k}{\nu}\|_2^2.$$

Let $f_k(\mathbf{w}) = \frac{\nu}{2}\|\mathbf{z}_k - \mathbf{m_q} + S_d\mathbf{w} - \frac{\boldsymbol{\mu}_k}{\nu}\|_2^2$. Using the second-order Taylor expansion, $f_k(\mathbf{w})$ is approximated as

$$f_k(\mathbf{w}) \approx f_k(\mathbf{w}_k) + (\mathbf{w} - \mathbf{w}_k)^\top \nabla f_k(\mathbf{w}_k) \\ + (\mathbf{w} - \mathbf{w}_k)^\top \mathbf{H}_f(\mathbf{w} - \mathbf{w}_k) \\ \approx f_k(\mathbf{w}_k) + (\mathbf{w} - \mathbf{w}_k)^\top \nabla f_k(\mathbf{w}_k) \\ + \psi\|\mathbf{w} - \mathbf{w}_k\|_2^2, \tag{10}$$

where the gradient of $f_k(\mathbf{w})$ at $\mathbf{w}_k$ is

$$\nabla f_k(\mathbf{w}_k) = \nu S_d^\top \left(\mathbf{z}_k - \mathbf{m_q} + S_d\mathbf{w} - \frac{\boldsymbol{\mu}_k}{\nu}\right), \tag{11}$$

and the Hessian matrix of $f_k(\mathbf{w})$ at $\mathbf{w}_k$ is $\mathbf{H}_f = \nu S_d^\top S_d$. Here, we approximate $\mathbf{H}_f \approx \psi\mathbf{I}$, and $\psi$ is determined by $\psi = \sqrt{\sum_i \sigma_i^2/N}$, where $\sigma_i$ are the i-th eigenvalues of $\mathbf{H}_f$. Using these approximations, we can rewrite (10) as

$$\mathbf{w}_{k+1} = \underset{\mathbf{w}}{\arg\min} \ \|\mathbf{w}\|_1 + \psi\|\mathbf{w} - \mathbf{w}_k + \frac{1}{2\psi}\nabla f_k(\mathbf{w}_k)\|_2^2. \tag{12}$$

According to [59], the optimal solution of (12) is

$$\mathbf{w}_{k+1} = soft\left(\mathbf{w}_k - \frac{1}{2\psi}\nabla f_k(\mathbf{w}_k), \psi\right), \tag{13}$$

where $soft(x, \psi) = \text{sign}(x)\max\{|x| - \psi, 0\}$. The solution for (8) is rather obvious, which is

$$\mathbf{z}_{k+1} = \frac{1}{2\lambda + \nu}(\boldsymbol{\mu}_k + \nu(S_d\mathbf{w}_{k+1} - \mathbf{m_q})). \tag{14}$$

Now we can summarize the ADM-based sparse optimization Algorithm 1. Consider the transition from the optimization problem $\mathcal{P}_1$ to $\mathcal{P}_2$. Given fixed vectors $\mathbf{m}_q$ and $S_d$, each $\epsilon$ corresponds to a unique $\lambda$, ensuring equivalence in the optimal solutions of the two problems. However, as $\mathbf{m}_q$ and $S_d$ dynamically change during testing, necessitating an adaptive relationship between $\epsilon$ and $\lambda$, it becomes imperative to employ an algorithm for the selection of a suitable $\lambda$. In this context, the binary search algorithm is employed to ascertain the optimal $\lambda$, as outlined in [27].

---

**Algorithm 1** ADM-Based Sparse Optimization

---

Initialize: $t = 0, \mathbf{w}_0 = 0, \mathbf{z}_0 = 0, \boldsymbol{\mu}_0 = 0, \lambda$
**while** not converged **do**
    Update the value of the $\mathbf{w}_{k+1}$ by equation (13).
    Update the value of the $\mathbf{z}_{k+1}$ by equation (14).
    Update the value of the $\boldsymbol{\mu}_{k+1}$ by equation (9).
    $\nu_{k+1} = 0.01\nu_k$ and $k = k + 1$.
**end while**
**return** $\mathbf{w}_{k+1}$

---

## 3.4 Few-Shot Image Recognition

In this work, we fuse the reconstruction similarity $P_R$ with the CLIP-based text-image similarity $P_{CLIP}$ for few-shot image recognition. Specifically, considering a specific class $d$, the scalar probability logit is computed as the negative mean squared Euclidean distance between $\mathbf{m_q}$ and $\mathbf{m_q^*}$ reconstructed from $\mathbf{S}_d$ across all feature map locations. It can be denoted as

$$\mathbf{m_q^*} = \mathbf{w_{k+1}} \mathbf{S}_d, \tag{15}$$

$$\langle \mathbf{m_q}, \mathbf{m_q^*} \rangle = \frac{1}{H_4 W_4} ||\mathbf{m_q} - \mathbf{m_q^*}||^2. \tag{16}$$

Consequently, For the reconstruction similarity $P_R$, the ultimate predicted probability is expressed as follows:

$$P_R(y_q = d|x_q) = \frac{\exp\left(-\epsilon \langle \mathbf{m_q}, \mathbf{m_q^*} \rangle\right)}{\sum_{d' \in D} \exp\left(-\epsilon \langle \mathbf{m_q}, \mathbf{m_q^*} \rangle\right)}. \tag{17}$$

Here, following [3, 50], we introduce a hyper-parameter $\epsilon$, denoted as a temperature factor. To obtain the CLIP-based text-image similarity $P_{CLIP}$, for the label of class $d \in D$, we place it in a manual prompt template such as "a photo of {class}", denoted as $\Pi_d$. We can obtain the text feature $f_t^d$ by $E_t$, denoted by $f_t^d = E_t(\Pi_d)$. First, we exploit $E_v$ to extract the global feature $\bar{\mathbf{m}}_\mathbf{q}$ of image $x_q$. Since both $\bar{\mathbf{m}}_\mathbf{q}$ and $f_t$ are $L2$-normalized, for the CLIP-based text-image similarity $P_{CLIP}$, the probability of $x_q$ belonging to class $d$ is:

$$P_{CLIP}(y_q = d|x_q) = \frac{\exp\left(sim\left(\bar{\mathbf{m}}_\mathbf{q}, f_t^d\right)/\tau\right)}{\sum_{d' \in D} \exp\left(sim\left(\bar{\mathbf{m}}_\mathbf{q}, f_t^{d'}\right)/\tau\right)}, \tag{18}$$

where $\tau$ is the learned temperature parameter of CLIP. $sim(\cdot, \cdot)$ denotes the following cosine similarity: $sim\left(\bar{\mathbf{m}}_\mathbf{q}, f_t^d\right) = \frac{\bar{\mathbf{m}}_\mathbf{q} \cdot f_t^d}{\|\bar{\mathbf{m}}_\mathbf{q}\| \|f_t^d\|}$. Finally, we fuse the inference from the visual representation reconstruction model and the original CLIP to obtain better predictions. The ultimate predicted probability of the input image $x_q$ is:

$$P_{total}(y_q = d|x_q) = P_{CLIP}(y_q = d|x_q) + \eta P_R(y_q = d|x_q), \tag{19}$$

where $\eta$ is used to control the scaling of the residual connection.

## 4 Experiments

### 4.1 Experimental Settings

For **few-shot recognition**, in adherence to established methods, our approach undergoes a few-shot evaluation across 11 widely employed image classification datasets. These datasets span a range of categories, encompassing generic object classification (such as ImageNet [38] and Caltech101 [11]), fine-grained object classification (including OxfordPets [35], StanfordCars [23], Flowers102 [34], Food-101 [1], FGVC Aircraft [31]), texture classification (represented by DTD [4]), remote sensing recognition (examined through EuroSAT [17]), scene recognition (explored in SUN397 [51]), and action recognition (evaluated on UCF101 [41]). Following CoOp [61], we test the **generalization** performance of our models from ImageNet to its variants: ImageNet-V2 [38], ImageNet-Sketch [49].

### 4.2 Implementation Details

Our method is built upon CLIP model, using ResNet-50 [16] as its image encoder and a transformer as its text encoder. Notably, both the visual and text encoders of CLIP remain frozen. We leverage prompt ensembling as defined in [36] and adhere to the data pre-processing protocol outlined in CLIP for all datasets. In Equation (19), we set the hyperparameter $\eta$ to 1.5 for ImageNet and 1.2 for the other 10 datasets. Our experimental design aligns with widely-used few-shot protocols, where random selections of 1, 2, 4, 8, and 16 examples per class are utilized for training, and subsequent evaluations are performed on the entire test set. For the domain generalization task, we directly utilize the model trained on 16-shot ImageNet to test its two variants individually. The penalty parameter $\nu$ is initially set to 1.5, and decays gradually by 0.01 to bring the solution closer to the optimal solution during iterations.

### 4.3 Performance Comparison

*(1) Training-Free Few-Shot Recognition.* We compare our method with the SOTA training-free methods: Tip-Adapter [56], Tip-X [46] and APE [63]. According to Figure 4, our Proposed TaCo outperforms all the baselines consistently and significantly from 1 to 16 shots, achieving leading performance among the methods for train-free few-shot recognition. Remarkably, we observe that TaCo achieves significant performance gain *i.e.*, +4.40% on FGVC Aircraft. Besides, our proposed TaCo maintains a distinctive performance on generic object classification with an accuracy gain of +1.88% on ImageNet, which demonstrates the efficacy of feature reconstruction with sparse optimization for few-shot recognition.

*(2) Incorporating TaCo with Existing Methods.* Since the optimization is performed on the feature map, Our method can be incorporated with existing methods such as CoOp [61], TaskRes [60], and PLOT [2]. In this paper, we conduct experiments on PLOT [2] combined with our Taco. Figure 5 presents the performance of TaCo when combined with other methods. Incorporated with prompt-based methods, our method consistently and significantly surpasses input-level prompting methods. Remarkably, on ImageNet, PLOT + TaCo with 1-shot outperforms bare PLOT with 16-shot. In comparison to feature-level adapter methods, our method still yields superior performance, outperforming them by a large margin. For example, with TaskRes involved, our method outperforms APE-T by up to 2.62% on 16-shot Food101 and 2.05% on 16-shot Standford-Cars. Overall, These results demonstrate the effectiveness of our Taco, and show the robust compatibility of our method, providing an immediate plug-and-play benefit to existing methods.

*(3) Domain Generalization.* The domain generalization setting assesses the model's ability to generalize to a target domain distinct from the source domain [44, 45]. We include seven previous methods encompassing zero-shot methods [15, 36], training-free methods [56, 63], and training-required methods [53, 56] for comparison. Our method consistently outperforms all the compared models across two out-of-distribution datasets by a large margin. In comparison to the second-best method, APE [63], TaCo outperforms it by up to 1.39% on ImageNet-V2. When combined with other style methods, our method exhibits distinctive generalization capability, exceeding APE-T [63] by 1.51% on ImageNet-V2. These results demonstrate the notable robustness of our method to shifts in distribution.

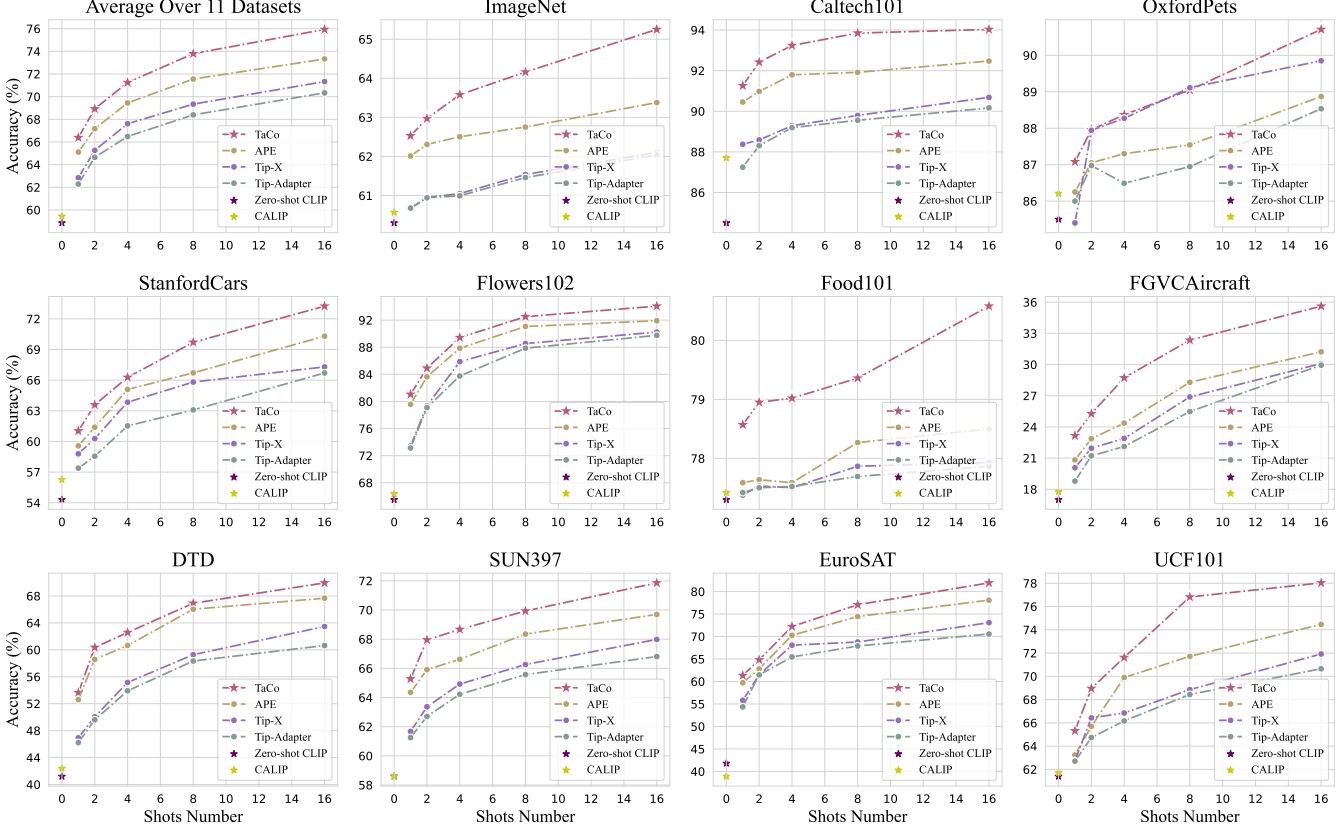

**Figure 4: Classification Performance Comparison on Training-free Few-shot Learning, *i.e.*, 1-/2-/4-/8-/16-shot, on 11 benchmark datasets. The top-left is the averaged accuracy over the 11 datasets.**

**Table 1: Performance comparisons on Domain Generalization.**

| Methods | Training Type | Source | Target | |
|---|---|---|---|---|
| | | ImageNet [5] | -V2 [5] | -Sketch [38] |
| CLIP [36] | Zero-shot | 60.33 | 53.27 | 35.44 |
| CALIP [15] | | 60.57 | 53.70 | 35.61 |
| Tip-Adapter [56] | Training-free | 62.03 | 54.60 | 35.90 |
| APE [63] | | 63.42 | 55.94 | 36.61 |
| TaCo (Ours) | | **65.25** | **57.33** | **38.07** |
| Tip-Adapter-F [56] | Training-required | 65.51 | 57.11 | 36.00 |
| TaskRes [53] | | 65.73 | 57.00 | 34.43 |
| APE-T [63] | | 66.07 | 57.59 | 36.36 |
| PLOT [2] + TaCo | | **67.13** | **58.62** | **37.03** |

**Table 2: Effectiveness of different algorithm components in TaCo. In this table, FMR represents Feature Map Reconstruction, and SO represents Sparse Optimization.**

| Method | Number of Shots | | | | |
|---|---|---|---|---|---|
| | 1 | 2 | 4 | 8 | 16 |
| Zero-shot CLIP [36] | 60.33 | 60.33 | 60.33 | 60.33 | 60.33 |
| CLIP + FMR(w/o SO) | 61.12 | 61.98 | 62.21 | 62.62 | 63.24 |
| CLIP + FMR + SO (Ours) | **62.53** | **62.97** | **63.58** | **64.16** | **65.25** |

optimization contribute significantly to performance improvement. Notably, in the 16-shot setting, our method using sparse optimization achieves a 2.01% performance gain. These results demonstrate the efficacy of sparse optimization in guiding feature reconstruction towards the most informative features, consequently yielding improved performance.

*(2) Evaluation on Various Visual Backbones.* Table 3 summarizes the results of 16-shot ImageNet [5] on various visual backbones. It can be observed that our method demonstrates substantial performance gains, particularly when compared to zero-shot CLIP on

## 4.4 Ablation Studies

In this section, we provide an empirical analysis of our design choices and the effects of different components of our method.

*(1) Contributions of Major Algorithm Components.* Our method is built upon CLIP, and we compare the different components integrated with CLIP across various shot settings. As shown in Table 2, our results indicate that both feature map reconstruction and sparse

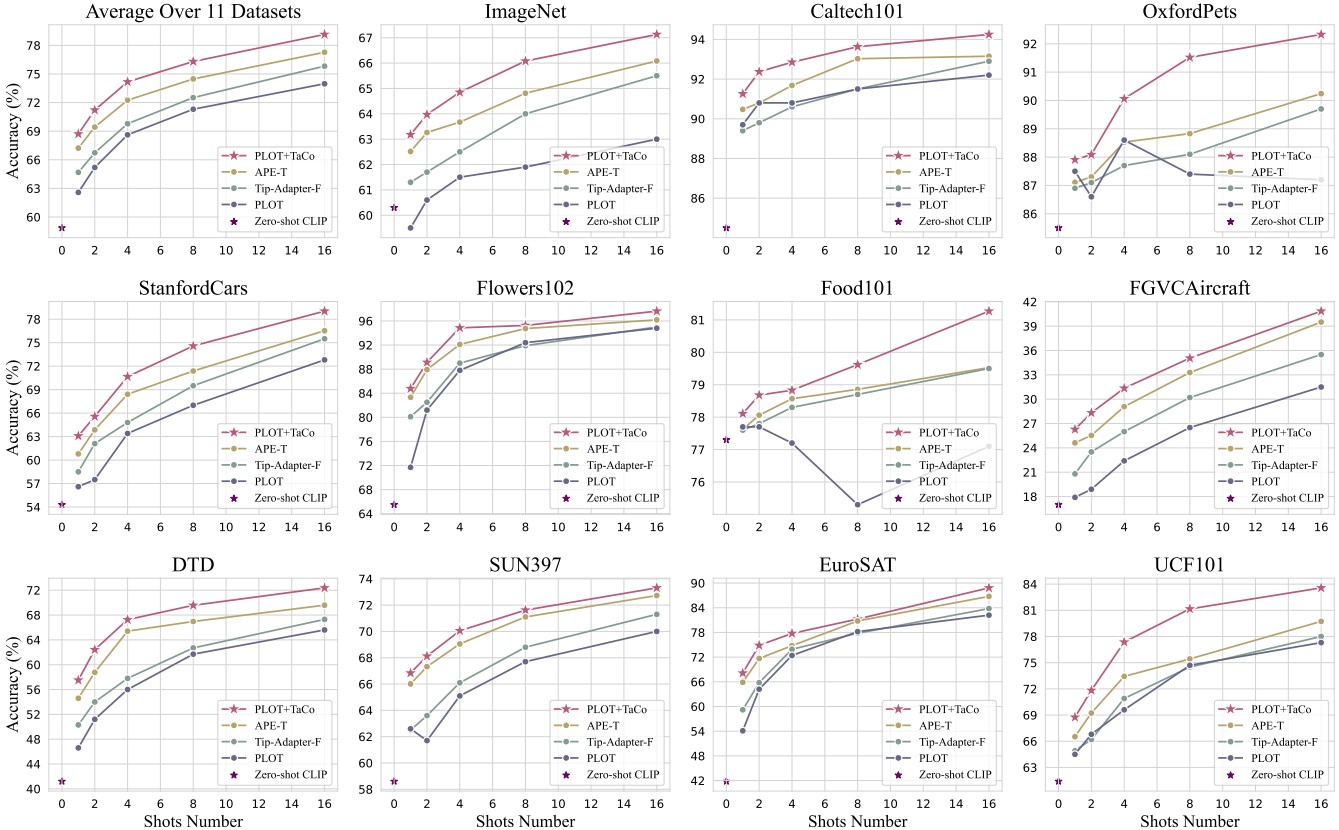

**Figure 5: Classification Performance Comparison on Training-required Few-shot Learning on 11 benchmark datasets.**

**Table 3: Evaluation across various visual backbones**

| Method | Visual Backbone | | | | |
|---|---|---|---|---|---|
| | ResNet-50 | ResNet-101 | ViT-B/32 | ViT-B/16 | ViT-L/14 |
| Zero-shot CLIP [36] | 60.33 | 62.53 | 63.80 | 67.83 | 75.43 |
| Tip-Adapter [56] | 62.03 | 64.78 | 65.61 | 70.75 | 76.19 |
| **Ours** | **65.25** | **66.34** | **68.12** | **72.85** | **79.57** |

**Table 4: Sensitivity of hyper-parameters. All the results are reported on a 16-shot setting on ImageNet [5].**

| Sensitivity of Hyper-parameters | | | | | | |
|---|---|---|---|---|---|---|
| $\eta$ | 0.0 | 0.5 | 1.0 | **1.5** | 2.0 | 2.5 |
| Acc. | 60.33 | 62.71 | 64.36 | **65.25** | 64.87 | 64.13 |

more advanced visual backbones. Our method shows consistent superiority against Tip-Adapter across all visual backbones.

*(3) Residual Ratio $\eta$.* The hyper-parameter $\eta$ controls how much to combine the predictions from feature reconstruction with pre-trained CLIP's prediction. This parameter can also be interpreted as weighing the reconstruction similarity in Equation (19). As formulated above, larger $\eta$ denotes depending more on reconstruction similarity and less otherwise. From Table 4, it is evident that the classification accuracy shows improvement as $\eta$ increases from 0.0 to 1.5, reaching its peak 65.25% at $\eta$ = 1.5. This observation suggests that reconstruction similarity contributes more than CLIP's text-image similarity regarding the final prediction. In the Supplemental Materials, we provide additional details of the proposed method and experimental results.

## 5 Conclusion

In this paper, we have studied the problem of adapting vision-language models (VLMs) to training-free few-shot image recognition. We formulated the few-shot image recognition task as a latent feature reconstruction and sparse optimization problem. Based on sparse optimization, we reconstruct the query feature from the corresponding support features and use the feature reconstruction error to formulate the reconstruction similarity. Coupled with the text-image similarity provided by the VLM, this reconstruction similarity analysis is able to accurately characterize the relationship between support and query images, thereby resulting in significantly improved performance in few-shot image recognition. Our comprehensive experimental results on few-shot recognition have demonstrated that the proposed method outperforms existing state-of-the-art approaches by large margins.

## Acknowledgments

This research was supported by the National Natural Science Foundation of China (No.62331014) and Project 2021JC02X103.

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
