# OpenReview forum: "Training-Free Feature Reconstruction with Sparse Optimization for Vision-Language Models"
_acmmm.org/ACMMM/2024/Conference — MM2024 Poster_

### Official Review · Reviewer_RxhA · 2024-05-19

**Rating:** 4
**Confidence:** 3

**Summary:**

The paper introduces training-free feature reconstruction with sparse optimizationn (TaCo), a novel approach for improving the performance of the adapting vision-language models in the training-free manner. TaCo harnesses the sparse optimization to reconstruct the query feature from the corresponding support features, then use the reconstruction error to define the reconstruction similarity. The experiments on different datasets demonstrate the effectiveness of proposed method.

**Strengths:**

1. The paper is easy to follow and it describes the proposed method in detail.
2. The optimization process is detailed.
3. The proposed method has good generalization performance and can be applied with different vision-language model backbones.
4. The proposed method achieves significant improvements in the performance of the adapting vision-language model, and outperforms the state-of-the-art methods in both the training-free and the training-required settings.

**Limitations:**

1. There is no time complexity analysis time of the Algorithm 1: ADM-Based Sparse Optimization, and the efficiency comparison, such as training time.
2. In the section 3.2, the discussion about "sparse optimization is beneficial for feature reconstruction" should be more specific and detailed.
3. There is a typo: $\text{S}_d^\mathsf{T}\text{w}$ should be $\text{S}_d^\mathsf{T}\text{w}_k$ at line 525.

**Suitability:**

2

---

### Official Review · Reviewer_u1TR · 2024-05-24

**Rating:** 3
**Confidence:** 3

**Summary:**

The paper introduces a training-free method for adapting vision-language models (VLMs) to few-shot image recognition. The method, called Training-free Feature ReConstruction with Sparse optimization (TaCo), addresses the challenge of effectively characterizing the semantic relationship between support and query samples without additional training. The authors propose using sparse optimization to reconstruct the query feature from support features in the semantic feature space, leveraging the observation that query and support images of the same class share a small set of distinctive visual attributes. This approach uses the feature reconstruction error to define reconstruction similarity to enhance few-shot image recognition performance. The method is evaluated on multiple few-shot recognition tasks, demonstrating substantial improvements over existing state-of-the-art approaches.

**Strengths:**

1. The paper introduces a training-free method that leverages sparse optimization for feature reconstruction in few-shot image recognition.

2. By utilizing the pre-trained VLMs without additional training, the method capitalizes on the robust visual and textual representations already present in models like CLIP. This training-free approach makes the method computationally efficient and easy to implement.

3. The paper provides extensive experimental results on various few-shot recognition benchmarks, including generic object classification, fine-grained classification, texture classification, and remote sensing recognition. The results consistently show that the proposed method outperforms several state-of-the-art approaches.

**Limitations:**

1. This paper only integrates VLMs to few-shot image recognition with spare optimization. The sparse optimization problem for query feature reconstruction based on an alternative direction is not novel.

2. How to avoid losing the significant feature on the sparse optimization of alternative direction.

3. Some typos need to be corrected. Such as line 163 lacks the ending symbol, which proves that this paper was written in haste and lacked careful correction.

**Suitability:**

2

---

### Official Review · Reviewer_gL15 · 2024-05-25

**Rating:** 5
**Confidence:** 3

**Summary:**

The paper presents a novel method, called Training-free Feature ReConstruction with Sparse optimization (TaCo), aimed at improving few-shot image recognition using vision-language models (VLMs) without the need for additional training. The core idea is to leverage sparse optimization techniques to reconstruct query image features from support image features, which are encoded using a pre-trained VLM. The reconstruction error is then used to measure the similarity between query and support images, supplemented by the text-image similarity provided by the VLM. This approach is designed to enhance the performance of few-shot learning tasks by accurately capturing the semantic relationships between images.

**Strengths:**

1. Novelty: The paper introduces an innovative method that combines sparse optimization with feature reconstruction in a training-free context for few-shot learning.  This approach addresses a significant gap in existing methods that struggle with characterizing semantic relationships without training.

2. Technical Correctness: The proposed method is grounded in solid theoretical concepts of sparse optimization and feature reconstruction. The mathematical formulation and optimization techniques are well-explained and justified.

3. Adequate Evaluation: The authors provide extensive experimental results demonstrating the superiority of their method over state-of-the-art approaches.  The evaluation covers various benchmarks and metrics relevant to few-shot image recognition, showcasing significant improvements.

4. Clarity: The paper is well-structured and clearly written.  The problem statement, proposed solution, and experimental results are presented in a logical and comprehensible manner.  Figures and diagrams effectively illustrate the key concepts and results.

**Limitations:**

1. Does the method assume that query and support images share a small set of distinctive visual attributes, which may not hold true in all cases, particularly in highly diverse or complex datasets, thus potentially limiting the method's effectiveness?
2. The experimental evaluation of the proposed method in the article primarily focus on image recognition tasks. Is there any applicability in other domains such as object detection or segmentation? Demonstrating versatility across different vision tasks would add significant value.
3. Can the code for this method be made publicly available?

**Suitability:**

2

---

### Official Review · Reviewer_FVY4 · 2024-05-28

**Rating:** 4
**Confidence:** 2

**Summary:**

This paper studies the task of few-shot image recognition without training, where support images containing certain entities are provided and the model is asked to recognize entities of the same class/type in the query image.

**Strengths:**

This paper is well-presented with detailed experimental results and accompanying analysis.

The evaluation is comprehensive, covering multiple datasets (eleven, in total) with ablations on e.g., different visual backbones.

**Limitations:**

**Motivation**

While the author's statement in Lines 131 and onwards is intuitive, specifically, the cat example, I am not entirely certain that the hypothesis derived from the example is theoretically sound. Specifically, I am referring to this particular sentence starting at Line 135:

> In the semantic feature space, the feature of the query image can be considered as a linear and sparse combination of support image features.

A similar sentence is also in the abstract Lines 13 and onwards. My concern is that the claim of the combination being *linear* and *sparse* may not stand, and essentially there is no concrete proof of such properties.

Granted, the results are impressive, but it does not indicate, in any way, that the original claim is sound. Given that the entire method is based on this hypothesis, it may bring ramifications to the foundation of this paper, which can be worrying.

As I am not necessarily an expert on the theoretical front, I will wait for other reviewers' input on this matter and adjust my recommendations accordingly.

**Note on suitability rating**

In the suitability rating below, I note that although this work utilizes vision-and-language models (VLMs), it does not directly develop, or train said models (as the method proposes a training-free paradigm for solving the task). Meanwhile, the studied task is not necessarily multi-modal in its nature.

To this end, I am rating it as *moderately suitable* for the community.

**Suitability:**

2

---

### Meta-Review · Area_Chair_bNSt · 2024-07-02

**Recommendation:** Accept (Poster)
**Confidence:** 5

**Metareview:**

The paper shows promise due to its innovative approach and strong experimental results. Addressing theoretical concerns, clarifying assumptions, demonstrating broader applicability, including efficiency analyses, and providing detailed discussions on the benefits of sparse optimization will strengthen the paper and better demonstrate its contributions.